# A Role for Superficial Heat Therapy in the Management of Non-Specific, Mild-to-Moderate Low Back Pain in Current Clinical Practice: A Narrative Review

**DOI:** 10.3390/life11080780

**Published:** 2021-08-02

**Authors:** Jürgen Freiwald, Alberto Magni, Pablo Fanlo-Mazas, Ema Paulino, Luís Sequeira de Medeiros, Biagio Moretti, Robert Schleip, Giuseppe Solarino

**Affiliations:** 1Department of Movement and Training Science, Bergische University Wuppertal, 42119 Wuppertal, Germany; freiwald@uni-wuppertal.de; 2S.I.M.G. Società Italiana di Medicina Generale, 50242 Florence, Italy; albertomagni@medicinsiemegarda.it; 3Faculty of Health Sciences, University of Zaragoza, 50009 Zaragoza, Spain; pfanlo@unizar.es; 4Farmácia Nuno Álvares, 2800-179 Almada, Portugal; ema.paulino@ezfy.eu; 5Nova Medical School, Nova University Lisbon, 1099-085 Lisbon, Portugal; luis.medeiros@chlc.min-saude.pt; 6Physical Medicine and Rehabilitation Department, Centro Hospitalar Universitário de Lisboa Central, 1150-199 Lisbon, Portugal; 7Orthopedic & Trauma Unit, Department of Basic Medical Sciences, Neuroscience and Sense Organs, School of Medicine, University of Bari Aldo Moro, 70124 Bari, Italy; biagio.moretti@uniba.it (B.M.); giuseppe.solarino@uniba.it (G.S.); 8Department of Sport and Health Sciences, Associate Professorship of Conservative and Rehabilitative Orthopedics, Technical University of Munich, 80992 Munich, Germany; 9Department for Medical Professions, Diploma University of Applied Sciences Bad Sooden-Allendorf, 37242 Bad Sooden-Allendorf, Germany

**Keywords:** heat therapy, low-back pain, musculoskeletal pain, non-pharmacological management

## Abstract

Low back pain (LBP) is a leading cause of disability. It significantly impacts the patient’s quality of life, limits their daily living activities, and reduces their work productivity. To reduce the burden of LBP, several pharmacological and non-pharmacological treatment options are available. This review summarizes the role of superficial heat therapy in the management of non-specific mild-to-moderate LBP. First, we outline the common causes of LBP, then discuss the general mechanisms of heat therapy on (LBP), and finally review the published evidence regarding the impact of superficial heat therapy in patients with acute or chronic non-specific LBP. This review demonstrates that continuous, low-level heat therapy provides pain relief, improves muscular strength, and increases flexibility. Therefore, this effective, safe, easy-to-use, and cost-effective non-pharmacological pain relief option is relevant for the management of non-specific mild or moderate low back pain in current clinical practice.

## 1. Introduction

Low back pain (LBP) is an exceptionally common musculoskeletal problem and a leading cause of disability [1]. LBP is experienced by most adults at some stage of their life, with an estimated 577 million people affected in 2017 [1]. A recent systematic literature review stated that the prevalence of LBP ranges from 1.4% to 20.0%, and the incidence from 0.024% to 7.0% [2]. LBP significantly impacts patients’ quality of life, limiting their daily activities and work productivity. Non-specific LPB therefore represents a substantial clinical and economic burden, and is a major public health concern [1,3,4,5,6,7].

LBP is usually self-limiting, resolving in one to two months in most patients. Treatments for non-specific LBP include both pharmacological and non-pharmacological interventions [8]. Although over-the-counter, non-steroidal, anti-inflammatory drugs are frequently used as a first-line treatment for LBP [9,10], they carry the potential risk of cardiovascular, renal, hepatic, and gastrointestinal complications, particularly when used longer term [11]. It should also be noted that LBP affects patients with different comorbidities, which may place limitations on the use of concomitant pharmacological therapies [12].

The current guidelines for patients with mild-to-moderate LBP that does not limit everyday activity recommend that patients self-treat or use alternative, non-pharmacological treatments, such as superficial heat, massage, acupuncture, or spinal manipulation as initial or complementary options [8,13]. Heat therapy has been used for centuries to relieve pain and promote health [14,15] and is applied today in a variety of forms, including heat pads or wraps, hot baths, and heat lamps [16,17]. These modalities act at different depths, with the collective action of reducing the muscle tone, increasing blood flow, and relieving pain [17,18]. Continuous, low-level heat therapy is an effective, easy-to-use, low-cost option that could be a valuable part of a multimodal analgesic strategy in current clinical practice, and is a useful treatment that patients can easily and safely self-administer [19].

This narrative review summarizes the role of superficial heat therapy for the management of both acute and chronic non-specific LBP, focusing on the use of continuous, low-level heat therapy administered directly to the skin via a heat wrap for patients with mild-to-moderate LBP based on published studies and our collective clinical experience.

## 2. Causes of Low Back Pain

LBP results from—among other factors—activation of nociceptors in response to trauma, tissue damage, or mechanical action on the spinal cord and spinal nerves, as well as changes in (inflammatory) metabolism. Specialized, group III or group IV free nerve endings (nociceptors) are polymodal [20,21], reacting to both mechanical influences and inflammatory processes. These free nerve endings can produce inflammatory mediators themselves (neurogenic inflammation amplification), which leads to a lowering of the receptor threshold and an amplification of pain (peripheral pain sensitization) beyond the primary cause [21].

“The changes in expression, distribution, and functioning of receptors and ionic channels are thought to be a part of the neuroplasticity property, through which the nervous system constantly adapts to external stimuli. Moreover, some of the reviewed mediators have also been associated with ‘central sensitization’, a process that results in pain chronification when the painful stimulation is particularly prolonged or intense, and lastly leads to the memorization of the uncomfortable painful perception” [22].

Neurotransmitters elicit changes in the interneurons of the spinal dorsal horns that influence their permeability, switching, and guidance mechanisms, which, in turn, affects the way they relay stimuli (including pain) [23,24,25,26,27]. Depending on their location and the cause of the activation of the pain receptors, there may be a local increase in the tone of the segmentally assigned muscles, with the formation of trigger points. A permanent change of the paravertebral muscle tone and the development of trigger points can lead to further pain intensification and can result in a vicious circle of pain amplification. In this context, Petrofsky et al. [28] showed that, when locally applied to trigger points, heat is significantly superior to sham treatment for non-specific neck pain.

Although the cause of LBP is often non-specific, extensional and rotational-shear forces acting on the spine can activate mechanical and polymodal pain receptors—especially in spinal segments with pre-existing damage. The activation of nociceptors alters neuromuscular activation, resulting in muscle inhibition and contributing to a degenerative cascade, which ultimately results in pain and limits the patient’s ability to move [29,30].

It is hypothesized that a low serum pH may drive LBP, as painful or damaged discs have a lower pH than non-painful discs. A more acidic environment stimulates the release of proinflammatory mediators and other inflammatory substances and depletes the proteoglycan within discs. In turn, proinflammatory cytokines drive the release of nerve growth factor, which facilitates the ingrowth of nerves into damaged discs and stimulates the production of pain mediators. Ultimately, these inflammatory pathways may alter the intricate nutrient balance of the nucleus pulposus (the inner core of the vertebral disc), leading to a reduced supply of oxygen, increased levels of lactate, and reduced pH, thereby continuing the cycle, further disrupting the disc microenvironment and exacerbating the pain stimulus [31,32].

Left untreated, a persistent stimulus can drive the transition from acute pain to chronic pain via a series of distinct pathophysiological steps. Persistent pain can activate secondary pathways that lead to peripheral and central sensitization, hindering normal functioning via long-lasting modification of the neuronal cytoarchitecture and loss of inhibitory interneurons [33,34]. It is therefore important that acute episodes of LBP are addressed in a timely manner, and within the “window” in which permanent changes may occur, to inhibit the transition to chronic pain [17,33]. Additionally, both chronic and recurrent LBP can be associated with changes in the structure and function of the paraspinal muscles (e.g., muscle degeneration and fat infiltration) that can compromise normal muscle biomechanics and restrict movement; however, discussion of these musculoskeletal changes is beyond the scope of this article.

## 3. Superficial Heat Therapy—How Does It Work?

The application of superficial heat is a non-pharmacological treatment approach that involves the application of a heat source to the body to raise the local tissue temperature (Figure 1). Heat therapy acts on pain and muscle spasms in multiple ways. The application of low-level superficial heat activates temperature-sensitive nerve endings (thermoreceptors), which, in turn, initiate signals that block the processing of pain signals (nociception) in the lumbar dorsal fascia and spinal cord [35]. In addition, the pressure used to apply some superficial heat therapies, such as heat wraps, may activate the nerve endings that detect changes in tissue pressure and movement (proprioceptors); when activated, the proprioceptors block the transmission of pain signals to the spinal cord and the brain. The analgesic effects of heat are partly mediated by transient receptor potential (TRP) membrane channels, of which seven respond to heat and two respond to cold temperatures. TRP vanilloid 1 (TRPV1) receptors facilitate the neural transduction of heat and the processing of nociceptive pain. The activation of TRVP1 receptors in the brain is thought to regulate anti-nociceptive pathways. These mechanisms serve to reduce muscle tonicity and relax muscles, thereby reducing spasms and musculoskeletal pain and increasing muscle flexibility [17,18,36].

In addition, an increase in temperature tends to reduce the stiffness in fascial tissues [37]. This effect may involve a decrease in the viscosity of hyaluronan, which restores normal gliding and normalizes the activity of the proprioceptive mechanoreceptors in the respective fascia [38]. An increase in thickness, together with reduced shear motion mobility of the thoracolumbar fascia in chronic low back pain, has been documented as a possible correlate of fascial scarring [39,40]. The application of heat in low back pain patients may therefore also involve a normalizing effect on the thoracolumbar fascia [37].

An increase in tissue temperature via the application of heat packs leads to increased metabolism and vasodilation and accelerates the healing processes. An elevation in tissue temperature of just 1 °C is associated with a 10–15% increase in the local metabolism. Heat-dependent vasodilation increases the blood flow at the site of injury, facilitating healing through an enhanced supply of nutrients and oxygen, and via the removal of pain-inducing mediators produced as a by-product of tissue damage. Connective tissues may also change in viscosity and density in response to heat, thereby improving the range of movement and enhancing tissue extensibility. Recent evidence also suggests that localized, repeated heat therapy may promote an angiogenic environment and enhance muscle strength [17,18,29,41,42].

## 4. Overview of Superficial Heat Therapy Modalities

Heat therapy is the therapeutic application of heat to the body that results in an increase in tissue temperature [17]. The mode of therapy can be superficial, delivered using conduction (e.g., heat wraps or heat packs) or convection (e.g., hydrotherapy) techniques, or deep, delivered by conversion methods (e.g., ultrasound, diathermy, and laser therapy). Table 1 summarizes the range of superficial low-level heat modalities available, all of which aim to provide pain relief and muscle relaxation/reduction in muscle spasms via the mechanisms described in the previous section [17,18]. Details on each of these heat therapy modalities and their specific applications are beyond the scope of this article, which is focused on the use of heat wrap therapy.

One advantage of superficial heat therapy is its safety profile. In a Cochrane review of nine studies, superficial heat therapy was associated with only minor adverse events, mostly in the form of “skin pinkness” that resolved quickly [16]. Despite this, the use of superficial heat therapy, especially at high temperatures, may carry the risk of burns or skin ulceration. Furthermore, in some specific causes of pain, it may cause disease complications, progression, or exacerbation of inflammation. Therefore, caution is required in any condition with sensory impairment, such as multiple sclerosis, spinal cord injuries, autoimmune diseases with joint pain, activated osteoarthritis, poor circulation, and cancer [43,44].

## 5. Evidence of the Effectiveness of Continuous Low-Level Heat Wrap Therapy for Low Back Pain

Before using heat therapy to treat LBP, it is important to rule out any serious systemic disease [45]. In a review of nine studies, including a total of 1117 patients, French et al. indicated that the continuous application of low-level heat directly to the skin via a heat wrap was shown to provide small, short-term improvements in pain and mobility [16]. Therefore, heat therapy represents a viable approach to the self-treatment of acute or chronic muscular LBP, either alone or as a part of a multimodal approach.

In a prospective study, patients with acute, non-specific LBP were randomized to receive continuous low-level heat wrap therapy for 8 h/day (*n* = 113), acetaminophen (*n* = 113), or ibuprofen (*n* = 106). Pain relief was reported in all three treatment groups, and was significantly greater with heat wrap therapy than with acetaminophen or ibuprofen throughout two days of treatment (*p* < 0.001 for all) and two days of follow-up (*p* < 0.001 for all). This translated into significant differences in pain relief scores of 33% for heat wrap vs. acetaminophen and 52% for heat wrap vs. ibuprofen. The heat wrap group also experienced significantly greater reductions in muscle stiffness and significantly greater improvements in lateral trunk flexibility and disability scores vs. the acetaminophen and ibuprofen groups after both the treatment and follow-up periods [46].

In another prospective trial, participants with acute LBP were randomized to receive heat wrap therapy (8 h/day for three consecutive days; *n* = 95) or oral placebo (*n* = 96). Heat wrap therapy resulted in significantly greater pain relief than oral placebo on Day 1 (*p* < 0.001; treatment difference of 68%); this extended to the end of the two days of follow-up (*p* < 0.0001). Pain relief correlated with other outcomes, in that a similar pattern was observed for muscle stiffness and lateral trunk flexibility, with significantly greater improvements in the heat wrap group vs. the oral placebo group on Day 1 through to the end of follow-up, as well as in disability scores from Day 3 [47]. A similar trial investigated the overnight use of a heat wrap in 76 patients with acute LBP. The overnight application (~8 h) of heat therapy for three nights was found to be significantly more effective at reducing pain during the following day and during the two days post-treatment than an oral placebo. Similarly, significant improvements were noted with a heat wrap vs. placebo for morning muscle stiffness, daytime muscle stiffness, pain–affect scores, and disability scores during treatment and follow-up. Furthermore, lateral trunk flexibility was significantly improved in the heat wrap group vs. in the placebo group at the end of treatment, and sleep scores were higher for the heat wrap group [48].

In two workplace studies, heat wrap therapy was found to significantly reduce pain intensity in patients with acute LBP, both during treatment and up to two weeks after its use [49,50]. Heat wrap therapy also reduced the impact of pain on everyday activities, most notably the ability to lift, work performance, and quality of sleep, and provided sufficient pain relief for most patients during treatment and two weeks after its use [49].

Heat wrap therapy has also been investigated as part of a multimodal approach in acute and chronic settings. In a prospective outpatient study that combined continuous heat wrap therapy with directional preference-based exercise, 100 patients with acute LBP were randomized to a heat wrap alone (~8 h/day for five days), exercise alone, a heat wrap plus exercise, or an educational booklet (control). Treatment lasted for five consecutive days, and patients were followed up for an additional two days. On Day 7, the functional improvement with a heat wrap plus exercise was 84%, 95%, and 175% higher than a heat wrap alone, exercise alone, or the control, respectively. By Day 7, the heat wrap plus exercise group had also achieved a significantly lower deficit from pre-injury function and a greater reduction in disability than all of the other treatment groups. Furthermore, the heat wrap plus exercise group was associated with significantly greater pain relief when compared with the exercise alone and control groups [51].

Although most studies of heat wrap therapy have examined its used in patients with acute LBP, heat wrap treatment has also been shown to have benefits in patients with chronic LBP. Within a multimodal approach trial of chronic LBP, 176 patients were randomized to receive basic multimodal treatment either alone or supplemented with heat wrap therapy. While range of movement and flexibility improved in both groups, after 12 weeks of treatment, the supplemented group recorded greater improvements in their strength parameters (extension and right/left rotation) than the non-supplemented group [29]. This supports previously reported short-term improvements in physical and psychological wellbeing associated with heat wrap therapy in the chronic setting [52]. The long-term effects of combining heat wrap therapy with exercise are currently under investigation for the management of acute LBP (ClinicalTrials.gov: NCT03986047). In this trial, patients will be randomly assigned to receive heat wrap therapy alone, exercise alone, or heat wrap plus exercise for seven days continuously, and will be followed for 24 weeks [53].

While most studies rely on subjective patient-reported scoring, Kettenmann et al. employed a spontaneous electroencephalogram (EEG) as an objective parameter alongside the self-assessment of pain [54]. Patients were randomized to oral analgesic as a rescue therapy (control; *n* = 15) or heat wrap therapy (≥4 h/day on four consecutive days) plus oral analgesic rescue therapy (*n* = 15). The heat wrap group had a shift of spontaneous EEG activity to lower frequency bands, indicating reduced arousal that was not observed in the control group. This is consistent with patient reporting, whereby heat therapy significantly reduced perceived pain vs. the control group. The heat wrap group also reported significant improvements in stress and quality of sleep [54].

In Table 2, we summarize some of the key trials that have examined the efficacy of continuous low-level (~40 °C) heat wrap treatment for the relief of acute or chronic LBP.

There were no serious adverse events reported in any of the aforementioned studies, and heat wrap therapy was found to be well tolerated in all of the studies mentioned [46,47,48,50,55].

Cost-effectiveness analyses have shown that the use of heat wrap therapy for the management of LBP is beneficial to both healthcare systems [56] and employers [49]. Economic modeling of the heat wrap vs. acetaminophen vs. ibuprofen study described above [46] indicates that introducing heat wrap therapy in place of oral treatments would provide material savings to the U.K.’s National Health Service [56]. Furthermore, a pharmacoeconomic analysis has demonstrated the improved workplace productivity, and subsequent benefit to employers, associated with heat wrap therapy [49].

## 6. Other Applications for Superficial, Low-Level Heat Therapy

The muscle relaxant and analgesic effects of superficial low-level heat therapy (as reviewed in the section above) have also been found to be efficacious in relieving other types of musculoskeletal pain. Several studies have reported the benefits of continuous, low-level, direct heat wrap therapy for the treatment of neck pain [28,57], knee pain (including pain from osteoarthritis, where a heat wrap was more effective than acetaminophen) [57,58,59,60], and wrist pain stemming from strain or sprain, tendinosis, and carpal tunnel syndrome, with particularly good results observed in patients with carpal tunnel syndrome [61].

Localized heating of certain trigger points has also proven effective at relieving neck pain; in this case, the heat is applied on the upper trapezius muscle [28]. Studies have also indicated that heat therapy is effective at preventing and treating delayed-onset muscle soreness associated with exercise, with benefits observed in younger and older patients, as well as those with diabetes (a group who reportedly experience greater muscle soreness after exercise) [57,62,63,64]. In addition, the application of heat therapy for 8 h, including the 4 h before exercise, was found to be significantly more effective than stretching at preventing pain and improving disability and physical function the day after exercise [62]. Further to this, studies have also indicated that heat therapy provides greater benefits than cold therapy when applied after exercise [62,63,64,65].

Heat wraps as a method of heat therapy for pain relief have shown the key advantage of wearability, which allows for continuous use and the rapid resumption of work/normal daily activities. This feature makes it particularly relevant for other areas of pain management, such as dysmenorrhea, where it has demonstrated pain relief comparable to that achieved with ibuprofen [66,67]. Heat therapy may also be beneficial as part of a long-term pain management strategy following some surgical procedures [68].

## 7. Conclusions

LBP exerts a substantial burden on patients and is recognized as a major public health concern. Early treatment can help to inhibit the transition from acute to chronic LBP. Several clinical trials have demonstrated that continuous, low-level heat therapy, used alone or as part of a multimodal approach, provides early pain relief and improves muscular strength and flexibility, facilitating a return to normal function in patients with either acute or chronic LBP. Although improvements seen with superficial heat therapy can be short-term, local heat therapy can have a valuable role in current clinical practice, particularly for complex clinical cases, such as elderly patients with multiple comorbidities who are already receiving several concomitant medications, and in the outpatient setting for the preparation and follow-up of back pain therapies. In patients with mild LBP, heat therapy may potentially negate the use of pain medications, and in patients with moderate-to-severe pain, heat therapy may help lower pain drug requirements (i.e., number and dose).

In conclusion, continuous, low-level superficial heat therapy is an effective, safe, easy-to-use, and cost-effective non-pharmacological pain relief option that patients can easily self-administer, proving that a therapy known for centuries still has a relevant role in clinical practice today.

## Figures and Tables

**Figure 1 life-11-00780-f001:**
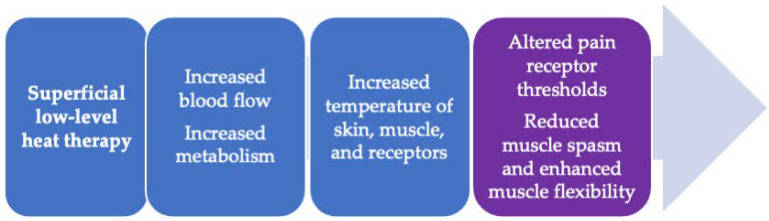
Mechanism of action of superficial heat therapy on low back pain (based on [17,18,29]).

**Table 1 life-11-00780-t001:** Types of superficial heat therapy used to treat acute or chronic back pain.

	Method of Heat Application	Type of Therapy
Superficial heat therapy	Conduction	Heat wrap (wearable) Heat pack (grain) Hot water bottles Hot poultices Hot stone therapy Electric heat pads
	Convection	Hydrotherapy Hot baths Heat lamp Stream/sauna

**Table 2 life-11-00780-t002:** Evidence for the effectiveness of heat wrap therapy in the treatment of low back pain.

Author (Year)	*N*	Study Design	Study Treatment	Comparator(s)	Primary Endpoint Results	Other Endpoints/Outcomes
Acute low back pain						
Nadler 2002 [46]	371	Prospective, randomized, single-blind, comparative, multicenter study Two days of treatment Two days of post-treatment follow-up	Continuous low-level heat wrap: 8 h/day at 40 °C for two days (*n* = 113)	Ibuprofen, 1200 mg/day (*n* = 106) Acetaminophen, 4000 mg/day (*n* = 113) Oral placebo (*n* = 20) Unheated back wrap (*n* = 19)	Pain relief on Day 1: Significantly greater with a heat wrap vs. acetaminophen (*p* = 0.0001) or ibuprofen (*p* = 0.0007)	Compared to acetaminophen or ibuprofen, a heat wrap wassignificantly associated with: (a) greater pain relief on Day 2 and extended pain relief (Days 3 and 4) (b) reduced muscle stiffness (Days 1–4) (c) improved flexibility (Days 2 and 4) (d) reduced disability (Days 2 and 4)
Nadler 2003 [47]	219	Prospective, randomized, parallel, single-blind, placebo-controlled, multicenter study Three days of treatment Two days of post-treatment follow-up	Continuous low-level heat wrap: 8 h/day at 40 °C for three consecutive days (*n* = 95)	Oral placebo (*n* = 96) Oral ibuprofen (*n* = 12) Unheated back wrap (*n* = 16)	Pain relief on Day 1: Significantly greater with a heat wrap vs. placebo (*p* < 0.001)	Compared to the placebo, a heat wrap was significantly associated with: (a) greater pain relief on Days 2 and 3 andextended pain relief (Days 4 and 5) (b) reduced muscle stiffness (Days 1–5) (c) improved flexibility (Days 1–5) (d) reduced disability (Days 3 and 5)
Nadler 2003 [48]	76	Prospective, randomized, parallel, single-blind, placebo-controlled, multicenter study Three nights of treatment Two days of post-treatment follow-up	Continuous low-level heat wrap: 8 h/night at 40 °C for three consecutive nights (*n* = 33)	Oral placebo (*n* = 34) Ibuprofen (*n* = 4) Unheated heat wrap (*n* = 5)	Morning pain relief on Days 2–4: Significantly greater with a heat wrap vs. placebo(*p* = 0.00005)	Compared to the placebo, a heat wrap was significantlyassociated with: (a) greater pain relief the following day and extended pain relief (Days 2–5) (b) reduced morning muscle stiffness in the morning and during the day (Days 2–5) (c) reduced disability at the end of treatment and follow-up (d) improved trunk flexibility on Day 4 (e) improved sleep quality and onset of sleep
Lurie-Luke 2003 [49]	52	Workplace intervention study Two days of treatment Two-week post-treatment follow-up	Continuous low-level heat wrap: 8 h/day at 40 °C for two consecutive days	–	A heat wrap significantly reduced pain intensity and impact of pain on work-related activities and sleep for two weeks post-treatment	A heat wrap was associated with a reduction in the use of other over-the-counter pain relief products A heat wrap was rated as “excellent” or “very good” by 44% of respondents and as “good” by 37%
Tao 2005 [50]	43	Randomized workplace study Three days of treatment 11 days of post-treatment follow-up	Continuous low-level heat wrap: 8 h/day at 40 °C for three consecutive days plus back pain education (*n* = 25)	Back pain education alone(*n* = 18)	Pain intensity and pain relief during treatment and follow-up: Heat wrap + education significantly reduced pain intensity (Days 1–14) and provided improved pain relief (Days 1–4) vs. education alone	Compared to education alone, heat wrap + education wasassociated with: reduced disability on Days 7 and 14
Mayer 2005 [51]	100	Randomized, controlled outpatient study Five days of treatment Two days of post-treatment follow-up	Continuous low-level heat wrap: 8 h/day at 40 °C plus exercise for five consecutive days *n* = 24)	Heat wrap alone (*n* = 25) Exercise alone (*n* = 25) Educational booklet (control; *n* = 26)	Functional ability: Heat wrap + exercise significantly improved functional outcomes vs. exercise alone (*p* = 0.18), or the control (*p* = 0.002) on Day 4 and vs. a heat wrap alone (*p* = 0.0007), exercise alone (*p* = 0.0003) or the control (*p* < 0.0001) at Day 7	Compared to a heat wrap alone, exercise alone, or the control, a heat wrap + exercise was significantly associated with: (a) less deficit from pre-injury function on Day 7 (b) reduced disability on Day 7 (and on Day 4 vs. the control) (c) greater pain relief on Days 4 (vs. the control) and 7 (vs. exercise alone and the control)
Kettenmann 2007 [54]	30	Randomized, active-controlled, parallel design study Four days of treatment One day of post-treatment follow-up	Continuous low-level heat wrap: ≥4 h/day at 40 °C for four consecutive days plus oral analgesics (as needed; *n* = 15)	Oral analgesics (as needed; *n* = 15)	Objective evidence of reduced pain arousal (EEG data): A heat wrap led to significantly greater drops in Beta-1 and -2 frequencies post-treatment vs. the control (Days 2 and 4)	Subjective evidence Compared to the control, a heat wrap was significantlyassociated with: (a) reduced pain (Days 2–4) (b) reduced stress (Day 3) (c) reduced tiredness (Days 2 and 4) (d) improved sleep quality (Day 4) (c) improved concentration (Days 2, 4) A heat wrap was rated as “excellent,” “very good,” or “good” by 86% of respondents
Stark 2014 [55]	61	Pilot study to evaluate sensitivity of two methods to assess time to onset of pain relief and flexibility	Continuous low-level heat wrap: 8 h at 40 °C (*n* = 26)	Oral placebo (*n* = 25) Sham wrap (*n* = 5) Oral ibuprofen (*n* = 5)	Median time to first pain and meaningful relief were both significantly shorter for heat wrap vs. placebo (*p* = 0.046 for both)	Compared to the placebo, a heat wrap was significantlyassociated with: (a) greater pain relief (b) greater change in muscle stiffness
Petrofsky 2015 [57]	145	Randomized, controlled outpatient study	Continuous low-level heat wrap: 6 h at 40 °C prior to home exercise program over two weeks (*n* = 71)	Home exercise program over two weeks without prior heat therapy (*n* = 7)	Compared to the control, a heat wrap was associated with a significantly greater: (a) improvement in strength after the two-week period (*p* < 0.01) (b) improvement in flexibility after the two-week period (*p* < 0.01) (c) reduction in disability after the two-week period (*p* < 0.01) (d) reduction in pain after the two-week period (*p* < 0.01) (e) compliance in the completion of home exercise (*p* < 0.01) Similar results were observed in patients with knee (*n* = 44) and neck (*n* = 59) pain	
Chronic low back pain						
Lewis 2012 [52]	24	Prospective single-arm study (within-subjects repeated measures design)	Continuous low-level heat wrap: 40 °C applied 2 h prior to assessment	Assessment without prior heat wrap application	Pain ratings were impacted by the fluctuating nature of chronic LBP Heat wrap treatment was associated with a reduction in non-normalized muscle activity and improved short-term well-being	
Freiwald 2018 [29]	176	Randomized, active controlled, multicenter, single-blind, observational study 12 weeks treatment	Continuous low-level heat wrap: 8 h at 40 °C plus multimodal treatment for 12 weeks (*n* = 88)	Multimodal treatment only (*n* = 88)	Muscular strength and flexibility: (a) strength and flexibility improved in both groups (b) significantly greater improvements in extension, and right and left rotation observed in the heat therapy-supplemented group	

## Data Availability

Not applicable.

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
