# Peer review of "A Role for Superficial Heat Therapy in the Management of Non-Specific, Mild-to-Moderate Low Back Pain in Current Clinical Practice: A Narrative Review"

_life, 2021, doi:10.3390/life11080780_

Round 1
Reviewer 1 Report
In this review, the authors analyse the use of heat therapy for the treatment of low back pain.
Even if this topic is quite interesting, some elements should be addressed in order to improve the manuscript:
It should be underlined that non-specific low back pain is the principal target of this therapy. I suggest also to add the term "non-specific low back pain" in the title of the manuscript.
The Authors in paragraph n. 2 analysed the different causes and mechanism of LBP. Even if some important elements are outlined, the peripheral and central sensitization mechanisms are reported only generically. Moreover, these elements are not key factor for their review. I suggest a more focused analysis of muscular component of LBP in this paragraph (e.g. Goubert D et al. Lumbar muscle structure and function in chronic versus recurrent low back pain: a cross-sectional study. Spine J. 2017 Sep;17(9):1285-1296) since muscles are the main targets of heat therapy.
The Authors cited different heat therapy’s mechanisms of action in Fig. 1. A mild increase in inflammation is reported in the figure but this element is not adequately reported in the manuscript with appropriate references.
The authors should clarify the role of heat therapy in acute vs chronic LBP and the short term improvements obtained with this technique.
The authors did not explain the criteria used in this review. How was literature searched? How many authors evaluated the literature? How papers were selected….? There are also excluded papers?
This review lacks the critical aspects needed in any review, please add your personal deduction based on literature, discusses it critically, identifies methodological problems and points out research gaps.
Moreover, I suggest you to highlight the limitations and strengths of the review.
Reviewer 2 Report
The authors present a paper on A role for heat therapy in low back pain in modern clinical practice whose main objective is to the role of heat therapy for the management of both acute and chronic non-specific LBP. However, they concentrate their entire study on, continuous, low-level modalities
The objective of the research is not clear, which has repercussions on the methodology.
Have no studies been found that also indicate that the deep modality improves the symptomatology of these ailments, or have the authors only focused on the low-mild modality?
If you have focused on the low modality, the title, abstract, body of the research and conclusion must be in line with the objective.
If you focus on the effects of heat in general, you should also detail the effects that are achieved with deep modalities
To further detail, based on scientific evidence, which of the superificial or deep techniques are the most advisable and why.
With respect to modern clinical practice ? what are the authors referring to?
The authors highlight this term in the title, so more emphasis should be placed on this part throughout the manuscript
Reviewer 3 Report
It was said that this study would be helpful in improving the understanding of the role of heat therapy for people with low back pain, but the treatment method related to heat therapy is too classic to be considered a new and fresh topic.
I think it should be possible to analyze meaningful results with more meaningful topics.
This paper deserves to be reject and invite resubmission.
Round 2
Reviewer 2 Report
Although the authors have only focused on the use of low-level, superficially/totally applied heat, with a greater emphasis on the clinical application of continuous low-level heat through thermal envelopes, However, the title does not reflect that they have only focused on surface or low level heat.
Regarding the body of the manuscript, point 3, although the objective is not a bibliographic analysis, it should be supported by this analysis to present/explain the heat treatment modalities. The analysis of the body of the manuscript needs more bibliographic support to back up what is presented, especially low level heat.
With respect to clinical practice, although the authors have changed the term modern clinical practice to clinical practice, they have not detailed the answer to the question, what does it refer to? Where has it been reflected in the text? As I commented, it should be emphasized in the manuscript.
Reviewer 3 Report
Accept in present form